# Boundary Matching and Interior Connectivity-Based Cluster Validity Anlysis

**Qi Li, Shihong Yue \*, Yaru Wang, Mingliang Ding, Jia Li and Zeying Wang**

School of Electrical and Information Engineering, Tianjin University, Tianjin 300072, China;
qili_2017@tju.edu.cn (Q.L.); yaruwang@tju.edu.cn (Y.W.); mlding@tju.edu.cn (M.D.); lijiajoyce@tju.edu.cn (J.L.);
wangzeying@tju.edu.cn (Z.W.)
\* Correspondence: shyue1999@tju.edu.cn; Tel.: +86-22-2740-5477

**Abstract:** The evaluation of clustering results plays an important role in clustering analysis. However, the existing validity indices are limited to a specific clustering algorithm, clustering parameter, and assumption in practice. In this paper, we propose a novel validity index to solve the above problems based on two complementary measures: boundary points matching and interior points connectivity. Firstly, when any clustering algorithm is performed on a dataset, we extract all boundary points for the dataset and its partitioned clusters using a nonparametric metric. The measure of boundary points matching is computed. Secondly, the interior points connectivity of both the dataset and all the partitioned clusters are measured. The proposed validity index can evaluate different clustering results on the dataset obtained from different clustering algorithms, which cannot be evaluated by the existing validity indices at all. Experimental results demonstrate that the proposed validity index can evaluate clustering results obtained by using an arbitrary clustering algorithm and find the optimal clustering parameters.

**Keywords:** clustering evaluation; clustering algorithm; cluster validity index; boundary point; interior point

---

## 1. Introduction

Clustering analysis is an unsupervised technique that can be used for finding the structure in a dataset [1–3]. The evaluation of clustering results plays a vital role in clustering analysis and usually is performed by a clustering validity index or several [4,5]. In the past decades, a large number of validity indices have been proposed to evaluate the clustering results and to determine the optimal number of clusters, which is an essential character of a dataset. These frequently used validity indices contain Davies-Bouldin measure [6], Tibshirani Gap statistics [7], Xie-Beni's separation measure [8], and etc. Moreover, the commonly used Bayesian Information Criterion (BIC) has been applied to estimate the number of clusters [9,10]. For example, [11] provided a closed-form BIC expression by imposing the multivariate Gaussian assumption on the distribution of the datasets. Then, a novel two-step cluster enumeration algorithm has been proposed by combining the cluster analysis problem. Thus, this new BIC method contains information about the dataset in both data-fidelity and penalty terms. Compared with the existing BIC-based cluster enumeration algorithms, the penalty term of the proposed criterion involves information about the actual number of clusters. Arbelaitz et al. [12] made comparisons among the existing validity indices. Recently, an unsupervised validity index [13] independent of any clustering algorithm was proposed for any dataset of spherical clusters. In addition, the idea has been proposed to deal with the clustering evaluation under the condition of big data [14,15], while very little work is available in the literature that discusses validity indices for big data. One of the few papers on

this topic that we are aware of is [16], but the method does not consider the complex data structures in today's computing environment.

However, the existing validity indices are greatly restrained by the following three disadvantages at least.

Specific algorithm. Usually, the existing validity indices can only evaluate the clustering results obtained by a specified algorithm (e.g., C-means [17,18] or Fuzzy C-means [19,20]) rather than an arbitrary clustering algorithm. If the clustering algorithm chosen for the dataset is not suitable, the evaluating results will not be guaranteed.

Specific parameter. Different parameters in a clustering algorithm lead to different clustering results. Most cluster validity indices aim to select the best one among all clustering results, and thus they regard the clustering parameter as their variable. These existing indices can only regard the number of clusters as a variable rather than other clustering parameters, such as the density threshold in Density-Based Spatial Clustering of Applications with Noise (DBSCAN) algorithm [21] and the grid size in the CLIQUE algorithm [22]. Recently, a density peak-point-based clustering (DPC) algorithm [23] and its variants [24–26] have attracted considerable attention; but the number of peak points therein remains so uncertain that the correctness of clustering results is difficult to guarantee.

Untapped result. A cluster consists of a high-density center and a group of relatively low-density neighbors around the center in DPC, a group of core points and corresponding boundary points around these core points in DBSCAN, a center and a group of points that are assigned to the cluster by the nearest neighbor principle in C-means. Consequently, all points in a cluster are partitioned into two types. An accurate clustering partition must result from the correct identification of the two types of points. Although the existing algorithms may partition all points into the two types of points, and the partitioning results fail to be taken into a validity index to evaluate the clustering results. Especially when several clustering algorithms are performed in the same dataset, it is impossible to choose the best clustering result.

To solve the above problems, we propose a nonparametric measure to find all the boundary [27,28] and interior points in any dataset at first. Moreover, once a clustering algorithm is performed, all the boundary and interior points in any cluster can be found. After the measurement of boundary points matching and interior points connectivity between the entire dataset and all partitioned clusters, a novel validity index is proposed. Three typical clustering algorithms, i.e., C-means, DBSCAN, and DPC, are applied to evaluate the generality of the novel validity index. Two groups of artificial and CT datasets with different characteristics validate the correctness and generalization of the proposed validity index.

## 2. Related Work

In this section, we will firstly review three typical clustering algorithms, and then discuss a group of mostly used validity indices.

### 2.1. Typical Clustering Algorithms

Assume that $X = \{x_1, x_2, \ldots, x_n\}$ is a dataset containing $n$ data points, and $S_1, S_2, \ldots, S_c$ are disjoint subsets of $X$. If the point $x_j$ belongs to the $i$-th subset $S_i$, then we set $u_{ij}$ equal to 1, or else 0. The binary membership function can be represented as follows

$$u_{ij} = \begin{cases} 1, & x_j \in S_i \\ 0, & x_j \notin S_i \end{cases} , \ i = 1, 2, \ldots, c; \ j = 1, 2, \ldots, n. \tag{1}$$

If each point belongs to one certain subset, then the partitioning of $X$ is called a hard partitioning, satisfying

$$X = S_1 \cup S_2 \cup \ldots \cup S_c, \ S_i \cap S_j = \phi, \ i \neq j, \ i, j = 1, 2, \ldots, c. \tag{2}$$

There is a great volume of clustering algorithms, but the following three algorithms in them are representative.

### 2.1.1. C-Means Algorithm

C-means has been widely used in almost all fields owing to its simplicity and high efficiency. Its detailed steps are listed in Algorithm 1.

---

**Algorithm 1. C-means Clustering Algorithm**

---

**Input:** the number of clusters $C$ and a dataset $X$ containing $n$ points.
**Output:** a set of $C$ clusters that contains all $n$ objects.
**Steps:**
1. Initialize cluster centers $v_1, v_2, \ldots, v_C$ by selecting $C$ points arbitrarily.
2. Repeat;
3. Assign each point to one certain cluster according to the nearest neighborhood principle and a chosen measure;
4. Update cluster centers by $v_i = \sum_{j=1}^{n} u_{ij} x_j / \sum_{j=1}^{n} u_{ij}$ for $i = 1 - C$;
5. Stop if a convergence criterion is satisfied;
6. Otherwise, go to Step 2.

---

The key parameter in C-means is the number of clusters ($C$) which has to be determined in the clustering process.

### 2.1.2. DBSCAN Algorithm

In comparison to the C-means, DBSCAN has two parameters: a neighborhood radius of any point and the number of points within the neighborhood. However, in practice, the two parameters have to be turned to a density measure, which usually is their rate and is used to find the density differences and distributed characteristics of all points. In terms of a density threshold, DBSCAN distinguishes core points and border ones from all points. If the density of a point is higher than $\varepsilon$, then this point is called a core point; or else a border point.

DBSCAN starts with an arbitrary point $p$ in $X$. If it is a core point, then assign all the points which are density-reachable from $p$ with $\varepsilon$ to the same cluster. If $p$ is not a core point, then $p$ is temporarily assigned to noise. Afterward, DBSCAN deals with the next point until all the points in $X$ have been visited.

The density threshold $\varepsilon$ in DBSCAN is a crucial parameter, and significantly affect the clustering results. As an example, a dataset with 176 points is evaluated (see Figure 1). Figure 1 shows the dataset is clustered into 5, 2, and 1 cluster with different values of $\varepsilon$, where points in different clusters are marked with different signs. It is clear that the number of clusters decreases as $\varepsilon$ increases. Therefore, if $\varepsilon$ is incorrectly chosen, then the number of clusters cannot be determined. Although Ordering Points to Identify the Clustering Structure (OPTICS) algorithm [29] can be applied to solve the value of $\varepsilon$, it neither provides clustering results explicitly nor is applied in high-dimensional data space.

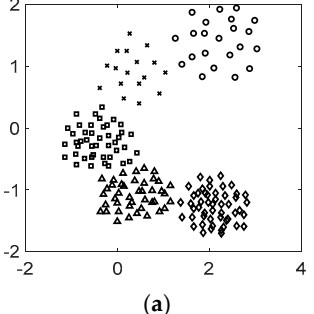 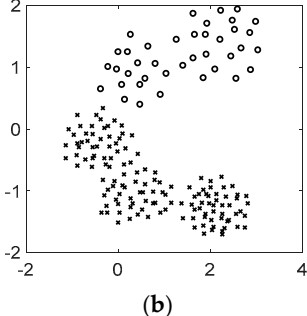 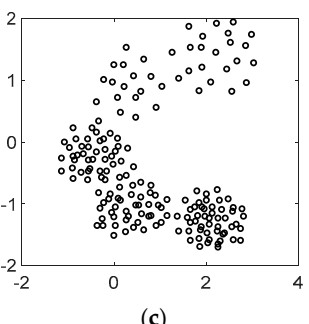

(**a**)          (**b**)          (**c**)

**Figure 1.** The number of clusters varies as $\varepsilon$ increases: (**a**) five clusters; (**b**) two clusters; (**c**) one cluster.

### 2.1.3. DPC Algorithm

DPC algorithm is the latest clustering algorithm, which combines the advantages of C-means and DBSCAN algorithms. For each point $x_i$, DPC calculates its density $\rho_i$ and its separated distance $\delta_i$ as follows.

$$\rho_i = \sum_j \chi(d(x_i, x_j) - d_c), \ s.t.\chi(x) = 1, \ if \ \chi(\bullet) < 0; \ else \ 0, \tag{3}$$

where $d_c$ is a cutoff radius and $d(x_i,x_j) = \|x_i - x_j\|$. $\delta_i$ can be measured by finding the minimum distance between the data point $x_i$ and other higher density points, i.e.,

$$\delta_i = \min_{j:\rho_j > \rho_i} d(x_i, x_j). \tag{4}$$

The cluster centers are these points that have relatively higher $\delta_i$ and $\rho_i$. After the calculation of $\gamma_i = \rho_i \delta_i$, $i = 1, 2, \dots, n$, the points with higher $\gamma_i$ are regarded as cluster centers. After determining the number of cluster centers, the clustering process can be used after scanning all points only once.

DPC, DBSCAN, and C-means have their own applicable ranges, respectively. Figure 2 shows three datasets with different characteristics: two nonspherical clusters in (a), three density-different clusters in (b), and three partially-overlapped clusters in (c), respectively. The clustering results by the above three algorithms are presented in these figures as well. Different numbers denote the real labels of the datasets, and different colors denote the clustering results by the corresponding clustering algorithms.

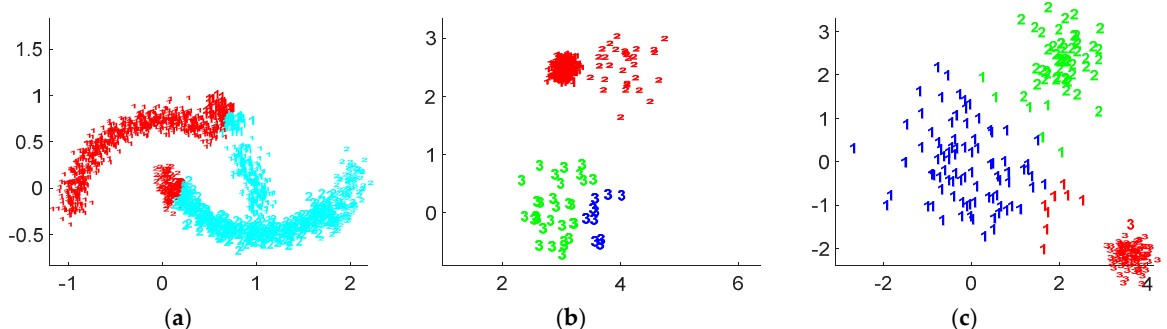

**Figure 2.** Three datasets with different characteristics and the clustering results by using density peak-point-based clustering (DPC), Density-Based Spatial Clustering of Applications with Noise (DBSCAN) and C-means: (**a**) two nonspherical clusters; (**b**) three density-different clusters; (**c**) three partially-overlapped clusters.

Figure 2a represents that DPC cannot work well when the investigated dataset contains a cluster with multiple density peaks. When one cluster has multiple peaks, DPC may regard these peaks as cluster centers, resulting in wrong clustering partition. Figure 2b shows the clustering results using the DBSCAN algorithm. In this case, there is none value of $\varepsilon$ to make DBSCAN find a correct clustering result. Figure 2c shows that the two clusters in the left are partially overlapped and have a similar density. C-means cannot find boundary points correctly, which leads to wrong clustering results.

When dealing with datasets in high dimensional (HD) spaces, these three clustering algorithms show different characteristics. Different from C-means and DPC, DBSCAN cannot effectively cluster points in HD datasets. First, the computed densities in DBSCAN have little difference among various high-dimensional (HD) points and cause that core points are difficultly determined. Accordingly, the merging condition to the same cluster from one to another core points has high uncertainty. Second, to accelerate DBSCAN, an R* tree structure indexing all points have to be used to decrease computational complexity, but it is challenging to be built in an HD space. Inversely, C-means depends on distance computation rather than density computation, and the distance differences among points can be much larger than their density differences. Also, DPC uses not only density $\rho$ but also separation

measure $\delta$ to find all abnormal points as cluster centers, and can significantly avoid the HD problem in DBSCAN. Consequently, the HD problem plays little effect on C-means and DPC.

C-means algorithm needs users to provide cluster number as the input parameter, which has an essential effect on clustering results. DPC has the same problem as C-means, which needs to provide the number of clusters in advance. The clustering process of DBSCAN has relevance with point density. Different values of neighborhood radius will result in different numbers of clusters, which can affect the clustering results.

Table 1 shows the characteristics of typical clustering algorithms when dealing with different types of datasets. The sign "×" denotes that the algorithm cannot cluster the corresponding types of datasets effectively according to the accuracy and applicable range, while "√" has the opposite meaning. The sign "√/×" means that the algorithm can obtain correct clustering results sometimes, while in some cases will not.

**Table 1.** Applicable range of typical clustering algorithms.

| Types/Algorithm | C-Means | DBSCAN | DPC |
|:---:|:---:|:---:|:---:|
| Arbitrary shape | × | √ | √/× |
| Density-diversity | √/× | × | √/× |
| overlap | √/× | × | √ |
| High-dimension | √ | × | √/× |
| The number of clusters | × | × | × |

### 2.2. Typical Cluster Validity Index

The validity index is a function, which regards the number of clusters ($c$) as its variable. This function can obtain its maximum or minimum value when $c$ is the correct number of clusters. It can be formulated as follows

$$\text{Max(min) } z = f(c), c = 1, 2, \dots, C. \tag{5}$$

The intra-cluster distances denote the compactness of a cluster while inter-cluster distances estimate the separation among clusters [30,31]. The trial-and-error strategy can be used to find the optimum solution in (5), as shown in Figure 3.

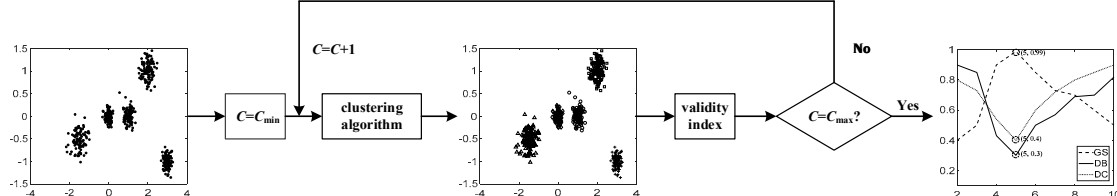

**Figure 3.** Evaluation processes of three existing validity indices.

Figure 2 and $c_{max} < \sqrt{n}$ [32] if there is no prior knowledge, where $n$ is the number of points in dataset $X$. Afterwards, apply an applicable clustering algorithm to $X$ with the value of $c$ set from $c_{min}$ to $c_{max}$. Calculate the corresponding value of (5). The maximum or minimum values of (5) indicates the optimal number of clusters. Note that different validity indices consist of different combinations of intra- and inter-cluster distances, and thus lead to different evaluation results.

In the following, three typical validity indices DB, GS, and, DC are illustrated. The evaluation process starts with $c_{min} = 2$, and ends with $c_{max}$ that is large enough. The maximums or minimums of the validity indices denote the optimal number of clusters, as explained below.

### 2.2.1. Davies–Bouldin (DB) Index

Let $\Delta_i$ and $z_i$ be the intra-cluster distance measure and cluster center of the $i$-th cluster, respectively. Let $\delta_{ij}$ denote the inter-cluster distance measure between clusters of $C_i$ and $C_j$, and $c$ can take values in $[c_{\min}, c_{\max}]$. The DB index [6] can be defined as

$$\mathrm{DB} = \sum_{i=1}^{c} R_i/c, \ s.t., R_i = \max_{j, j \neq i}(\Delta_i + \Delta_j)/\delta_{ij}, \ \delta_{ij} = \|z_i - z_j\|, \ \Delta_i = \sum_{x \in C_i} \|x - z_i\|/|C_i|. \quad (6)$$

### 2.2.2. Dual-Center (DC) Index

For any clustering center $v_i$ determined by a partitional clustering algorithm, assume $\dot{v}_i$ is the closest prototype to $v_i$, then the dual center is calculated as $\ddot{v}_i = (v_i + \dot{v}_i)/2$. Finally, a novel dual-center (DC) index [33] can be constructed, i.e.,

$$\mathrm{DC}_c = \sum_{i=1}^{c} \Delta_i(c) / \sum_{i=1}^{c} \delta_i(c), \ s.t., \Delta_i = \sum_{j=1}^{n_i(c)} (x_j - v_i)^2, \ \delta_i = \sum_{j=1}^{\ddot{n}_i(c)} (x_j - \ddot{v}_i)^2, \quad (7)$$

where $n_i(c)$ and $\ddot{n}_i(c)$ are the number of points of the $i$-th cluster when the prototypes are regarded as $v_i$ and $\ddot{v}_i$, respectively. Among the existing validity indices, DC has higher accuracy and robustness when dealing with both artificial datasets and real datasets in UCI [34].

### 2.2.3. Gap Statistic (GS) Index

The gap statistic (GS) index [7] firstly computes an intra-cluster measure as

$$W_c = \sum_{i=1}^{c} D_i/(2|C_i|), s.t., D_i = 2|C_i| \sum_{j \in C_i} \|x_j - \bar{x}\|, \bar{x} = \sum_{i=1}^{|C_i|} x_i/|C_i|. \quad (8)$$

Owing to the subjectivity of the detection of inflection point, GS can be formulated as

$$Gap_c = E^*[\log(W_c)] - \log(W_c), \text{ and } W_c = \sum_{i=1}^{c} D_i/(2|C_i|), \quad (9)$$

where $E^*$ denotes the expectation under a null reference distribution.

In sum, the above validity indices all take a trial-and-error way for a single specified clustering algorithm rather than general clustering algorithms. Moreover, these validity indices are the function of the number of clusters and are not designed for other possible clustering parameters. Therefore, an efficient and comprehensive method is necessary, which can evaluate clustering results for any clustering algorithm and arbitrary clustering parameters. In this paper, our proposed validity index presents an accurate solution to solve the above problem in a general way.

## 3. Materials and Methods

In a nonparametric way, we firstly partition all points into two groups, boundary and interior points, which are used to access the boundary matching degree and connectivity degree. By integrating these two quantities, a novel clustering evaluation index can be formed.

### 3.1. Boundary Matching and Connectivity

The density of any point in the existing clustering analysis is computed by counting the number of points in the point's neighborhood with a specified radius. However, the computed density only is a group of discrete integers such as 1, 2, $\ldots$ , and thus many points have the same density which is indistinguishable. Moreover, the defined density may greatly be affected by the specified radius.

In this study, we first define a nonparametric density to find all boundary and interior points in any dataset. Assume $X = \{x_1, x_2, \ldots, x_n\}$ is a dataset in a $D$-dimensional space $R^D$. For any data point $x_k \in X$, its $m$ nearest neighbors are denoted as, $x_{k,1}, x_{k,2}, \ldots, x_{k,m}$, with distances $d(x_k, x_{k,1}), d(x_k, x_{k,2}), \ldots, d(x_k, x_{k,m})$, where $m$ is the integer part of $2D\pi$, $k = 1, 2, \ldots, n$. Here, $2D$ shows that one

interval in any dimension in $R^D$ can be measured by the two-interval endpoints, and $\pi$ is a conversing coefficient when the $m$ points are enclosed by a spherical neighborhood that is used in the existing density computation. Therefore, the density of any data point $x_k$ in $X$ is defined as

$$density(x_k) = \left\{ \sum_{j=1}^{m} d(x_k, x_{k,j}) \right\}^{-1}, \ k = 1, 2, \ldots, n \tag{10}$$

**Definition 1.** *Boundary point and interior point. A point is called as a boundary or interior point if half of its m nearest neighbors have a higher or lower (equal) density than its density, respectively.*

The proposed notion of boundary and interior points have the following two characteristics.

(1) Certainty. Unlike the used density in other existing algorithms, the proposed density is fixed and unique for any point, which reduces the uncertainty in the clustering process. In fact, the clustering results in other algorithms may greatly be changed as the number of neighbors used for computing the density increases or decreases. Note that the effective estimation of the number of neighbors has been a difficult task, and so far, it remains unsolved [35].

(2) Locality. The classification of border or interior points is defined only by its $m$ nearest neighbors, so the separating boundary or interior points presents local characteristics. Inversely, DBSCAN uses a global density to distinguish border or interior points in a density-skewed dataset, which even causes an entire cluster to be perfectly regarded as border points (see Figure 4). Therefore, the proposed density is a more reasonable local notion.

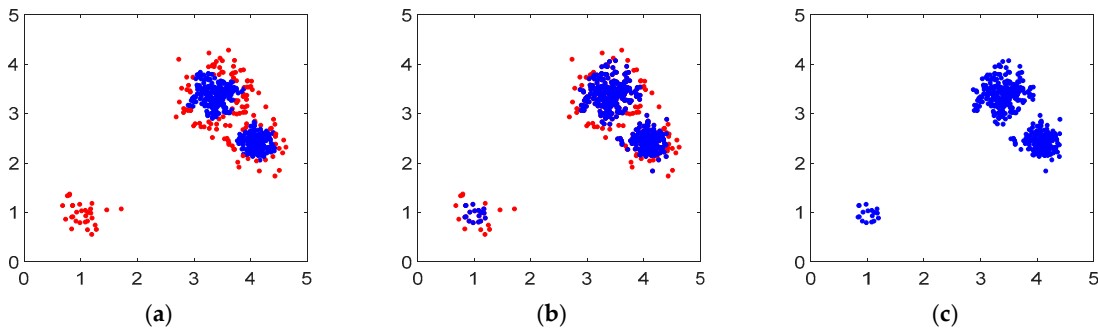

**Figure 4.** Comparison of boundary points determined by DBSCAN and (10): (**a**) global density; (**b**) local density; (**c**) interior points.

Figure 4a shows a density-skewed dataset with three clusters of large, medium, and small density, respectively. The red points in Figure 4a,b represent the boundary points computed by using DBSCAN and (10), respectively. DBSCAN can find no interior points in the cluster with the lowest density; in comparison, owing to the locality of (10), the red border points and blue interior points determined are distributed more reasonably. The interior points are located at the center of any cluster and surrounded by border points, while the border points construct the shape and structure of any clusters. Specifically, after the removal of border points in any dataset, the separation of clusters is greatly enhanced (see Figure 4c). Therefore, the real number of clusters can be determined more easily by any clustering algorithm.

In graph theory, a cluster is defined as a group of points that connect to each other [36,37]. In order to assess the connectivity of points in a dataset $X$, we calculate the density for all points in $X$ and sort them in the order of increasing density. Assume $x_{max}$ is the point with the highest density in $X$, and thereby, a connecting rule among points is defined as follows. For any point $x_{k \in X}$, the next point $x_{k+1}$ is the point which is the nearest neighbor of $x_k$ but has a higher density than $x_k$. Subsequantly, repeat the above steps until visiting the point $x_{max}$.

**Definition 2.** *Chain. A chain is a subset of X that starts with any data point $x_i$ in X and stops at $x_{max}$ based on the above connecting rule.*

There is a unique chain from any point in $X$ since the nearest neighbor of each point is unique. The above steps are repeated until each point has been visited in $X$. Consequently, all $n$ points in $X$ respond to $n$ chains, denoting them as $S_1, S_2, \ldots, S_n$. The largest distance between adjacent points in $t$-th chain $S_t$ is denoted as $dis(S_t)$, $t = 1, 2, \ldots, n$. Figure 5 shows all chains in two datasets with various characteristics, where the arrow is the direction from low to high-density points. The green dotted circles denote the points with maximum densities.

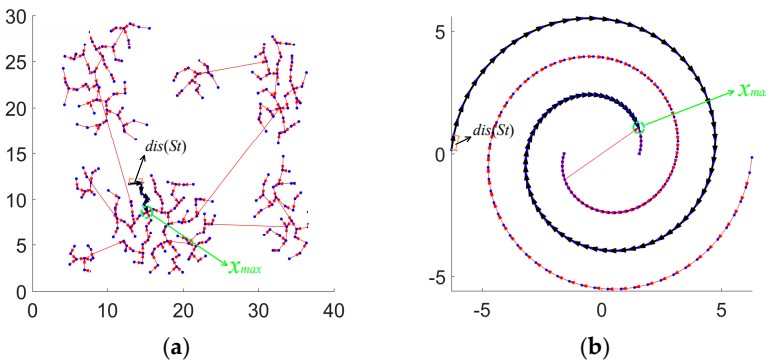

| (a) | (b) |

**Figure 5.** All chains in two datasets with different characteristics: (**a**) chains in a dataset of spherical clusters; (**b**) chains in a dataset of nonspherical clusters.

Figure 5 shows that the value of $dis(S_t)$ is small when a chain perfectly is contained in a cluster, but abnormally becomes large when a chain bridges a cluster and the other cluster. In views of the notation of the chain, we further define a notation of connectivity of $X$ as follows.

**Definition 3.** *Connectivity. Let $S_1, S_2, \ldots, S_n$ be n chains in X, then the connectivity of all points in X is defined as*

$$con(X) = \left( \sum_{t=1}^{n} dis(S_t) \right)^{-1}. \tag{11}$$

Along with the graph theory, the value of $con(X)$ indicates the degree of compactness of a dataset. It can reflect whether a chain is contained in a cluster, as explained and illustrated in the next section. In this paper, we use the notion of boundary matching degree and connectivity degree to access the clustering results obtained by any clustering algorithm.

*3.2. Clustering Evaluation Based on Boundary and Interior Points*

Once a clustering algorithm has partitioned a dataset $X$ into $c$ disjoint clusters, i.e., $X = C_1 \cup C_2 \cup \ldots \cup C_c$, we substitute $X$ by $C_1, C_2, \ldots, C_c$, respectively, and find their boundary points using (10). The set of boundary points in $C_k$ is denoted as $BC_k$, while the set of boundary points in $X$ is $BX$. A boundary-point-matching index is defined as

$$bou(c) = \sum_{k=1}^{c} \frac{|BX \cap BC_k|}{|BX \cup BC_k|}. \tag{12}$$

Equation (12) measures the matching degree of boundary points between the entire dataset $X$ and the disjoint $C$ clusters. In the mathematical meaning, it is clear that the values of $bou$ ($c$) must fall in the interval [0, 1]. The following example can explain the cases that are smaller and equivalent to 1.

Figure 6a–c show the boundary and interior points determined by (10) when performing C-means algorithm at $c$ is smaller, equal to or larger than 6, respectively, where interior points refer to the remaining points after removing boundary points.

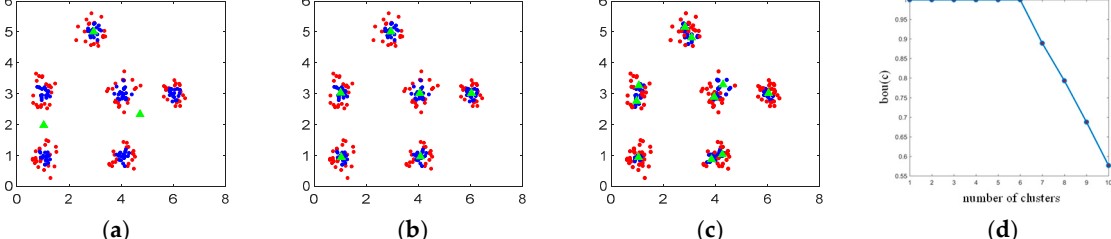

**Figure 6.** (**a**–**c**) shows the determined boundary points by using C-means and (10) when $c$ = 3, 6, and 10, respectively. Blue and red points refer to the interior and boundary points, respectively; (**d**) shows the curve of *bou* ($c$) calculated according to (11); the sign $\Delta$ means the center of C-means.

The red boundary points at $c$ = 3 and 6 are similar to these in $X$, but these boundary points at $c$ equals 10 are different from those in $X$. Figure 6d shows that the values of *bou* ($c$) are nearly unchangeable when $c < 6$ but decrease fast when $c > 6$. When the number of clusters $c$ is smaller than the actual one, and usually, any cluster does not be assigned two cluster centers. Therefore, the set of boundary points of all partitioned clusters are consistent with the entire set $X$, and *bou* ($c$) = 1. When $c$ is larger than the actual number of clusters, there is at least one cluster the number of whose boundary points increase. Thus, *bou* ($c$) < 1. It can be seen that the values of *bou* ($c$) are helpful to find the real number of clusters for any clustering algorithm. Alternatively, we can regard any cluster in $C_1, C_2, \dots ,$ $C_c$ as an independent dataset like $X$, and accordingly, $x_{max}$ in $X$ become these points with maximal density in $C_1, C_2, \dots ,$ and $C_c$, respectively. Thereby, we assess the connectivity among points according to (11) when $c = 1, 2, \dots , c_{max}$. The connectivity of $X$ is reduced to the connectivity of each cluster. As $c$ increases, the connectivity is enhanced since the number of maximal inter-cluster distances in these chains decreases.

Figure 7a–c show the connectivity calculated using (11) and C-means when $c$ equals 3, 6, and 10, respectively. This value becomes smaller when $c < 6$, while it tends to be flat when $c > 6$, as shown in Figure 7d. Consequently, there is an inflection point on the curve in Figure 7d.

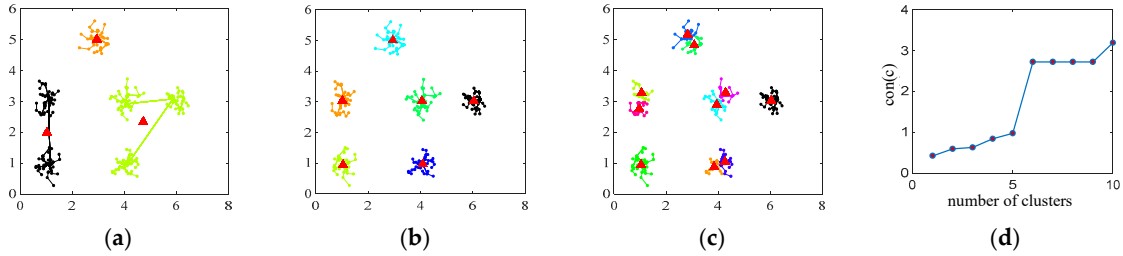

**Figure 7.** (**a**–**c**) The chains when c = 3, 6, and 10, respectively; (**d**) shows the curve of (11). Note that the points in the same color indicate that they are partitioned into the same cluster by using C-means algorithm and the sign $\Delta$ means the center of C-means.

As $c$ increases, both the curves calculated according to (11) and (12) have inversely varying tendencies. It is expected that the real number of clusters is encountered at $c^*$, where the curve of (11) turns to be flat from fast-changing, and that of (12) becomes fast varying from slow-changing.

Considering the variances of *bou* ($c$) and *con*($c$) can be calculated by curvature radius mathematically, we define a novel validity index according to *bou* ($c$) for boundary points and *con*($c$) for interior points, respectively. By combining (11) and (12), we define a function as follows

$$F(c) = R_1(c) \bullet R_2(c)$$

$$s.t., \begin{cases} R_1(c) = \left|\Delta_1(c)\right|^2 / \left(1 + (\nabla_1(c))^2\right)^{3/2} \\ \Delta_1(c) = bou(c+1) + bou(c-1) - bou(c), \nabla_1(c) = bou(c+1) - bou(c-1) \\ R_2(c) = \left|\Delta_2(c)\right|^2 / \left(1 + (\nabla_2(c))^2\right)^{3/2} \\ \Delta_2(c) = con(c+1) + con(c-1) - con(c), \nabla_2(c) = con(c+1) - con(c-1) \end{cases} \qquad (13)$$

where the symbol $\Delta$ denotes a two-order difference operator of $bou(c)$ and $con(c)$, aiming to locate the maximal inflection points on curves of $bou(c)$ and $con(c)$, respectively. The optimal number of clusters $c^*$ for any dataset is computed as

$$c^* = \text{argmax}_c F(c). \qquad (14)$$

The proposed validity index has the following characteristics.

(1)  Complementarity. The mathematical curvature and difference can reflect the varied tendency of a curve in (13), and thereby the real number of clusters $c^*$ can be found. When $c < c^*$, $R_1(c)$ is nearly equivalent to $R_1(c^*)$ since the set of boundary points is approximately unchangeable, but $R_2(c) < R_2(c^*)$, since the number of center points successively increase. In sum, $F(c) < F(c^*)$. Inversely, when $c > c^*$, $R_1(c)$ successively decreases, and $R_2(c)$ tends to be flat; therefore, $F(c) < F(c^*)$. Consequently, (13) can attain a maximum when $c^*$ appears.

(2)  Monotonicity. Assume that $c^*$ is the real number of clusters, and when $c$ takes its two values $c_1$ and $c_2$ satisfying $c_1 < c_2 < c^*$, $F(c_1) < F(c_2) < F(c^*)$; otherwise, when $c^* < c_1 < c_2$, $F(c^*) > F(c_1) > F(c_2)$; Hence, $F(c)$ consists of two monotone functions at the two sides of $c^*$, respectively. Therefore, for arbitrary two values $c_1$ and $c_2$ satisfying $c_1 < c_2$, $c_2$ may refer to a more optimal clustering result. Usually, a larger value of $F(c^*)$ indicates a better clustering result.

(3)  Generalization. Equation (13) can provide a wide entry for any clustering algorithm only if the clustering results are available and the corresponding numbers of clusters are taken as the variable of $F(c)$. Especially, a group of clustering results may result from different clustering algorithms and parameters, since any two clustering results are comparable according to the above monotonicity. For example, one clustering result with $c_1$ results from C-means, and others from DBSCAN, and so on. Equation (13) can evaluate the results of any clustering algorithm and parameter in a trial-and-error way. In comparison, the existing validity indices can mainly work for a specific algorithm and parameter of the number of clusters since the center in them has to be defined, especially for the C-means algorithm.

Hereafter, the cluster validity index of (13) based on boundary and interior points is called CVIBI. The evaluating process for any clustering results based on CVIBI is listed in Algorithm 2.

---

**Algorithm 2. Evaluating Process Based on CVIBI**

---

**Input:** a dataset $X \in R^D$ containing $n$ points and clustering results from any clustering algorithm at $c = 1, 2, \ldots, c_{max}$.

**Output:** the suggested number of clusters.

**Steps:**

1. Calculate the density for each point in $X$ according to (10);
2. Partition $X$ into boundary and interior points;
3. Input clustering results at $c = 1, 2, \ldots, c_{max}$;
4. Partition each cluster into boundary and interior points;
5. Compute values of $bou(c)$ or $con(c)$ at $c$ equals $1, 2, \ldots, c_{max}$;
6. Solve the optimal value of (13);
7. Suggest an optimal number of clusters.
8. Stop.

---

## 4. Results and Discussion

We test the accuracy of CVIBI on two groups of typical datasets and compare it with three existing validity indices, i.e., DB, DC, and GS. In views of different characteristics of the investigated datasets, the clustering results are obtained using C-means, DPC, and DBSCAN algorithms, respectively, where the number of neighbors $m$ in the experiments is fixed at the integer part of $2D\pi$.

### 4.1. Tests on Synthetic Datasets

Figure 8 shows five groups of synthetic datasets generated by the Matlab® toolbox, and each group consists of three datasets with different numbers of clusters. The determined boundary and interior points are in red and blue, respectively. The 15 datasets are denoted as sets 1–15, respectively. groups 1–4 contain the datasets with various densities, sizes, shapes, and distributions, respectively; and group 5 contains overlapped clusters.

The characteristics of these datasets are listed in Table 2. The first column of Table 2 denotes the names of these datasets. The second and fourth columns represent the numbers of clusters and objects of datasets, respectively. The third column denotes the dimensions of these datasets. The last column shows the number of objects of each cluster in datasets.

**Table 2.** Characteristics of 15 datasets in Figure 8.

| Name | Clusters | Dimension | Number of Objects | Number of Each Cluster |
|---|---|---|---|---|
| *Set* 1 | 3 | 2 | 600 | 83/164/353 |
| *Set* 2 | 4 | 2 | 350 | 30/60/120/240 |
| *Set* 3 | 5 | 2 | 830 | 30/60/120/240/480 |
| *Set* 4 | 3 | 2 | 420 | 60/120/240 |
| *Set* 5 | 4 | 2 | 900 | 60/120/240/480 |
| *Set* 6 | 5 | 2 | 1860 | 60/120/240/480/960 |
| *Set* 7 | 3 | 2 | 1018 | 341/336/341 |
| *Set* 8 | 4 | 2 | 404 | 134/90/90/90 |
| *Set* 9 | 5 | 2 | 494 | 134/90/90/90/90 |
| *Set* 10 | 3 | 2 | 360 | 120/120/120 |
| *Set* 11 | 4 | 2 | 800 | 200/200/200/200 |
| *Set* 12 | 5 | 2 | 1000 | 200/200/200/200/200 |
| *Set* 13 | 3 | 2 | 600 | 200/200/200 |
| *Set* 14 | 4 | 2 | 400 | 100/100/100/100 |
| *Set* 15 | 5 | 2 | 1000 | 200/200/200/200/200 |

The number of neighbors $m$ plays an essential role in CVIBI. Different values of $m$ may lead to very different density and distance values, and thus to different evaluation results.

As an example, we solve the range of $m$ in which each number can lead to the correct number of clusters when using DPC. Let $m = 1, 2, \ldots, n$, respectively. Along with these values of $m$, DPC is used to cluster all points in sets 1–15, respectively. Figure 9 shows the solved ranges in the 15 datasets that are represented by blue bars, where the red line in any bar denotes the value that is the integer part of $2D\pi$. All the experimental datasets used in Figure 8 are in the two-dimensional data space. Thus, $D = 2$ and the integer part of $2D\pi$ is just 12. Figure 9 shows that all the values of $2D\pi$ fall into the solved ranges corresponding to the correct number of clusters, which demonstrates the effectiveness and robustness of $2D\pi$. Consequently, we can fix the number of neighbors $m$ at $2D\pi$ in experiments.

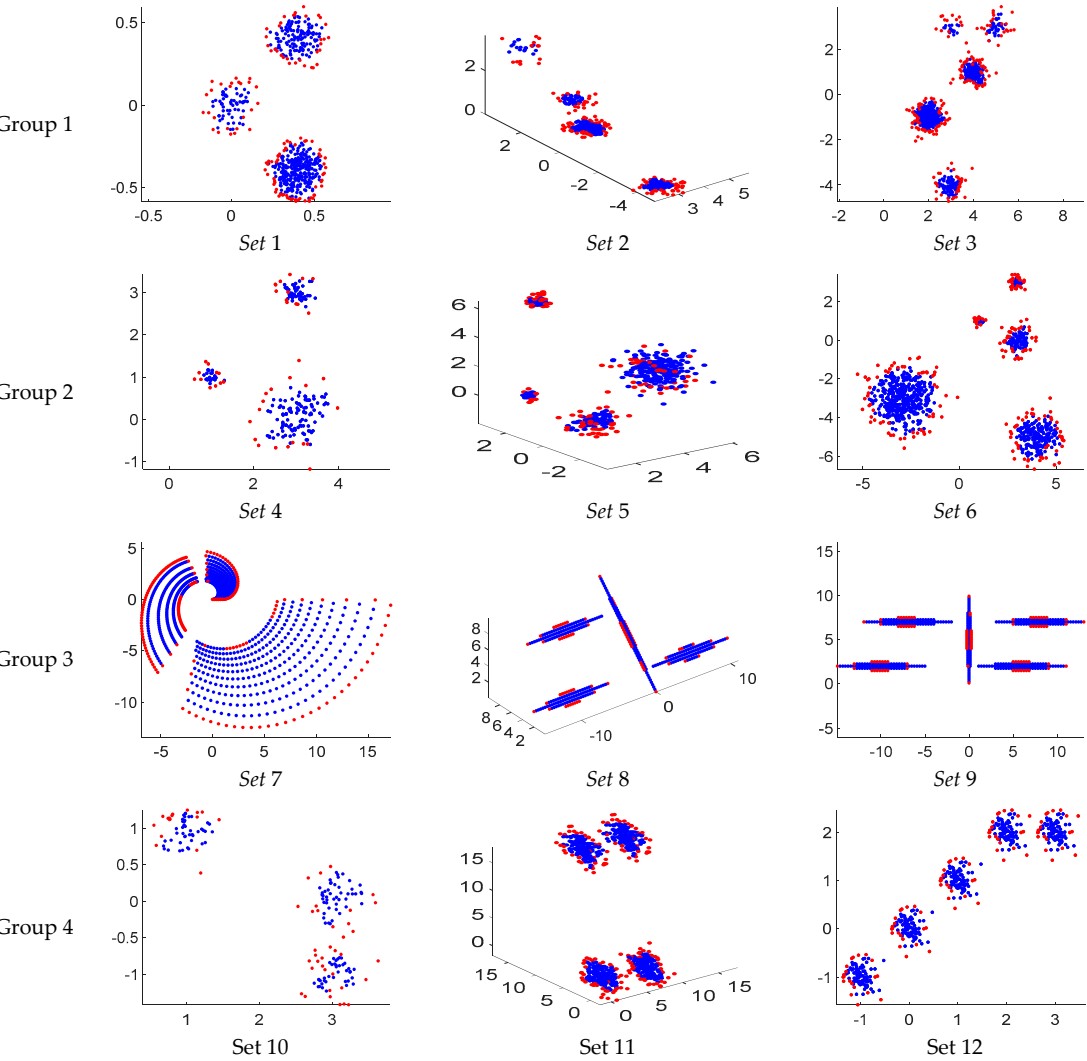

**Figure 8.** *Cont.*

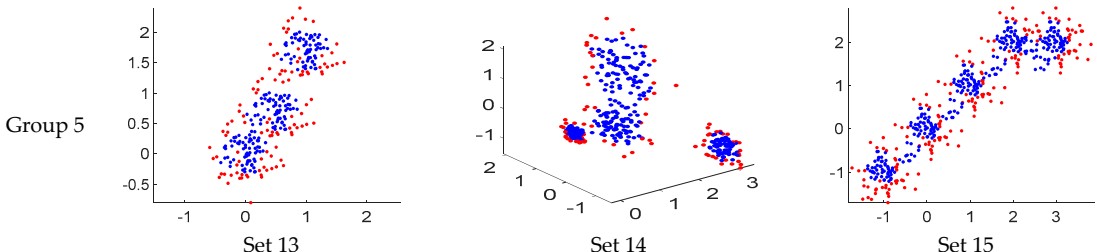

**Figure 8.** Five groups of synthetic datasets.

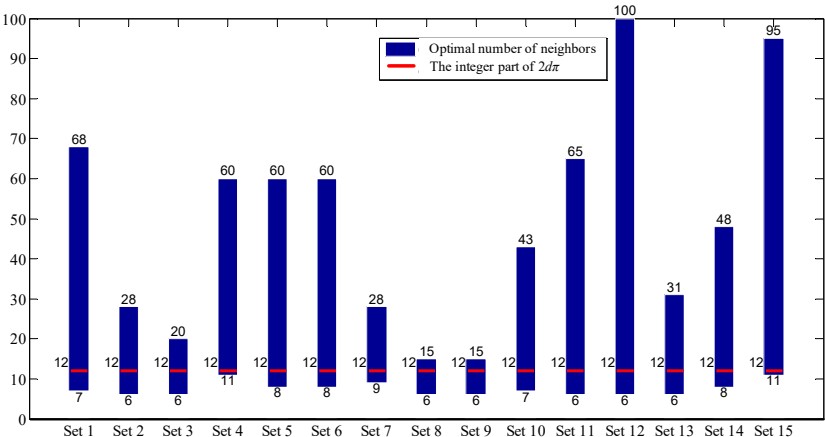

**Figure 9.** The solved range of the number of neighbors in 15 datasets.

### 4.1.1. Relationship between CVIBI and the Number of Clusters

The correctness of clustering evaluation depends on whether the correct number of clusters can be found by the validity index. In order to test the correctness of CVIBI, all points in these 15 datasets are partitioned into $c$ clusters by DPC, C-means, and DBSCAN at $c = 1, 2, \ldots, c_{max}$, respectively, where the number of clusters in DBSCAN is obtained by taking various values of $\varepsilon$. Figure 10 shows the curves of CVIBI based on the three algorithms, respectively. The points marked by small circles in these curves are the suggested optimal values of (13).

Datasets in group 1 and group 2 contain clusters with different densities and sizes, respectively. CVIBI can point out these optimal clustering results on the above six datasets no matter what clustering algorithms are used. In terms of various shapes in *Group* 3, CVIBI works well based on DPC and DBSCAN, but C-means cannot. Because C-means originally is designed to partition spherical clusters rather than arbitrary shapes, and then incorrect partitions of boundary points and incorrect values of *con* ($c$) are caused. When the clusters in datasets contain different distributions (e.g., group 4), CVIBI based on DPC shows better performance than C-means and DBSCAN according to accuracy. Because C-means and DBSCAN cannot give correct clustering results, DPC can obtain relatively correct results. When the dataset has clusters that partially overlap with each other (e.g., group 5), DPC can obtain relatively correct clustering results than C-means and DBSCAN, as shown by the corresponding curves. The merit of CVIBI is to select the best one from any candidates of clustering results, no matter what clustering algorithm is used. However, if all available clustering results cannot contain the real number of clusters, CVIBI can find it as well.

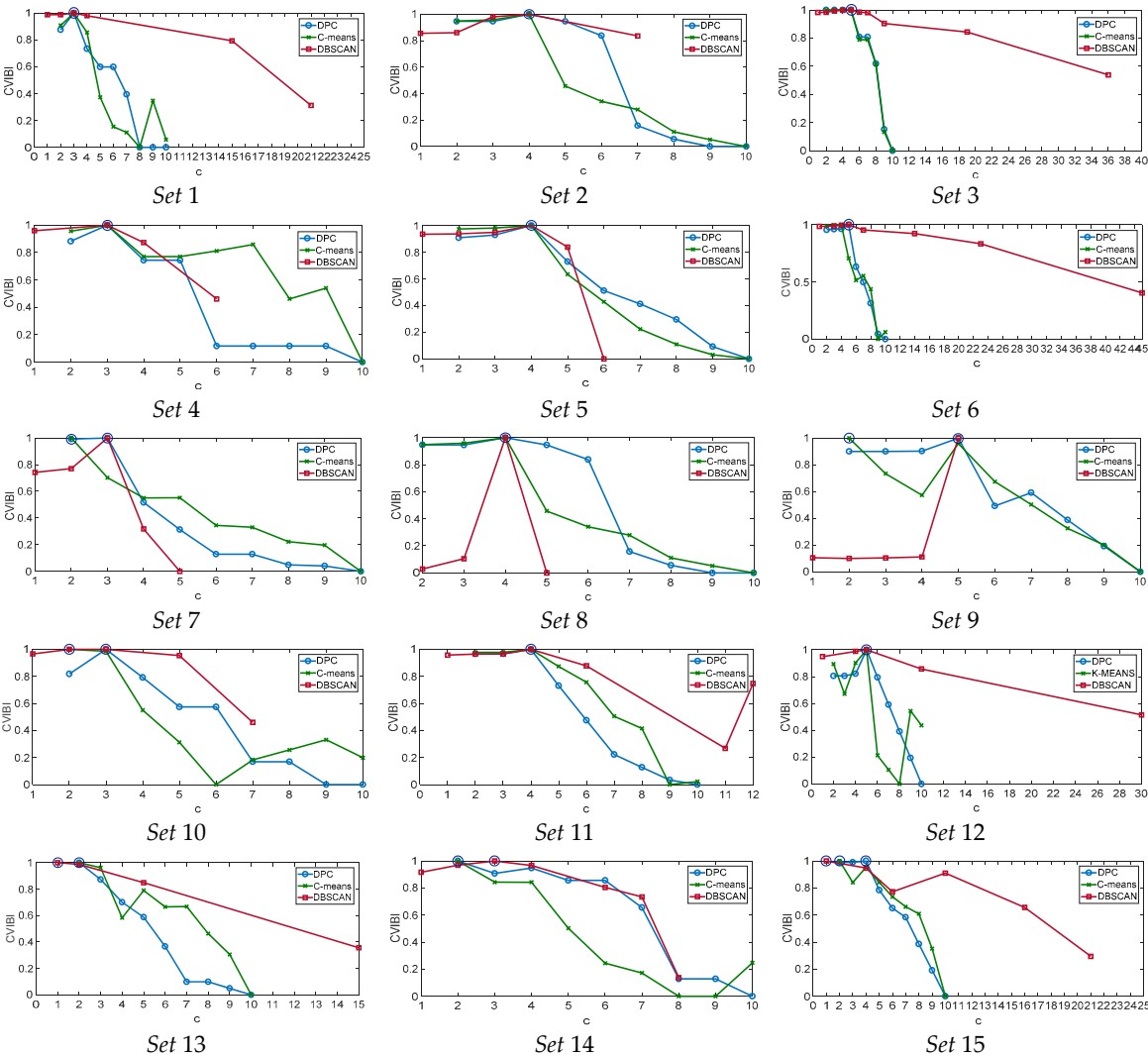

**Figure 10.** Evaluation results of CVIBI obtained using DPC, C-means, and DBSCAN.

### 4.1.2. Relationship between CVIBI and $\varepsilon$ in DBSCAN

When using DBSCAN, any certain number of clusters responds to a continuous interval of *Eps* (i.e., $\varepsilon$). The clustering results of DBSCAN highly depend on the value of *Eps*. In order to solve the optimal values of *Eps*, we determine the possible range of *Eps* first. Then, a group of *Eps* values is evenly sampled to approximate the *Eps* variable itself. Then, we cluster the dataset using DBSCAN with the group of *Eps* values. Finally, the optimal values of *Eps* can be determined by CVIBI. Figure 11 shows the evaluation results of CVIBI with respect to *Eps*, where piecewise numbers chain in these CVIBI curves is the obtained number of clusters. The abscissa and vertical axes denote the value of *Eps* and CVIBI, respectively. The number along with each red point denotes the number of clusters computed by DBSCAN. If CVIBI can compute the real number of clusters, then a green box will be drawn and point out the optimal value of *Eps*. The maximum value of each curve in Figure 11 indicates the suggested optimal number of clusters. The corresponding values of *Eps* are the suggested optimal parameters. It also can be seen that the optimal *Eps* is not one certain number but a range of values. CVIBI with DBSCAN can point out the correct clustering results except *Set* 10 and group 5. Set 10 has three clusters and one cluster is separated from the other two clusters. The other two clusters are close to each other, which are easy to be treated as one cluster. CVIBI with DBSCAN algorithm regards these two clusters as one cluster, and the maximum value occurs when $c = 2$. The clusters in group 5 are

significantly overlapped, so CVIBI cannot recognize the real numbers of clusters. Consequently, CVIBI can point out the optimal clustering results by finding the optimal parameter of *Eps*.

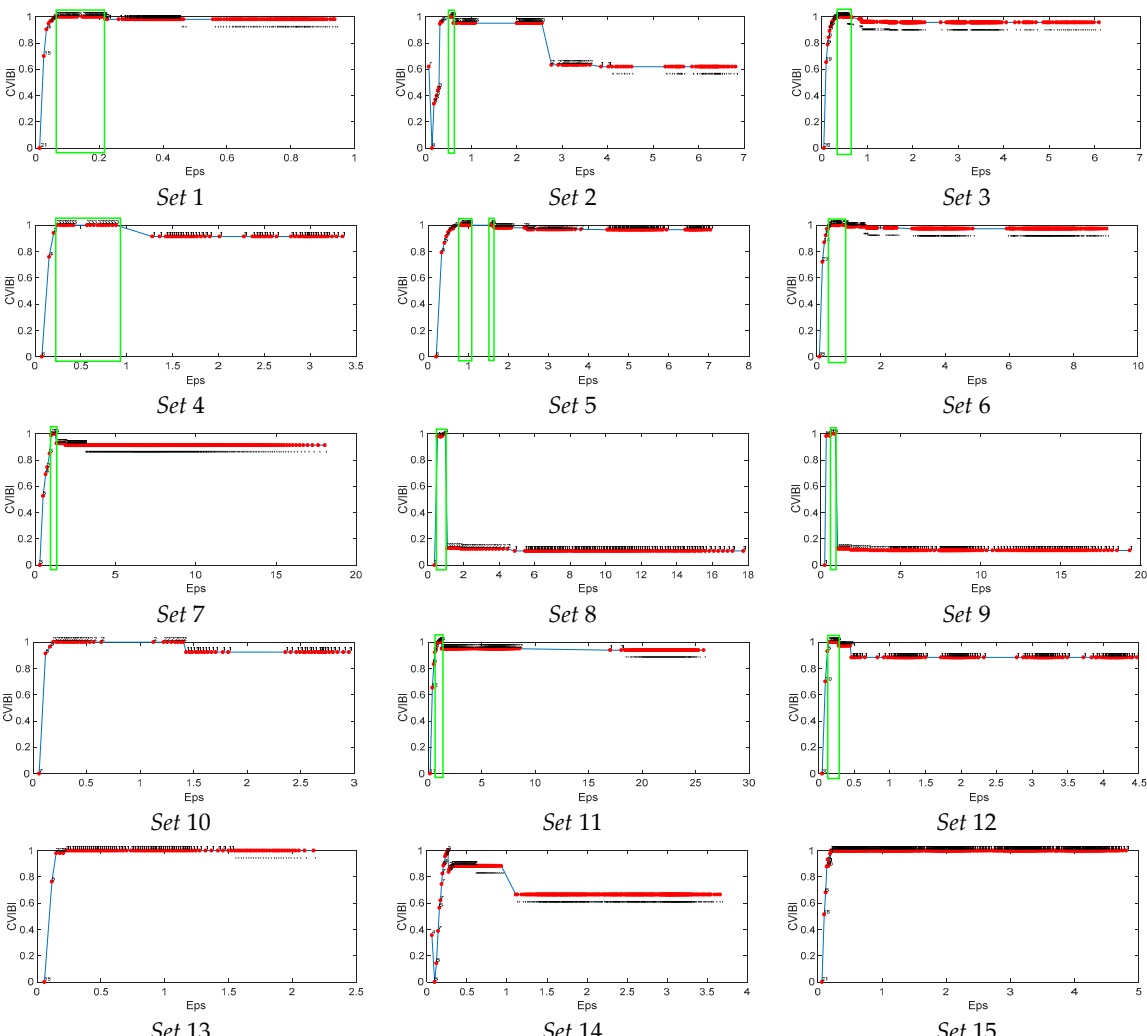

**Figure 11.** Evaluation results of CVIBI based on DBSCAN.

### 4.1.3. CVIBI Evaluation under Various Clustering Algorithms

Once a dataset is clustered by several clustering algorithms that lead to different clustering results, how one can select the optimal clustering result remain a challenging task for the existing validity indices. Nevertheless, CVIBI can realize this purpose. For example, CVIBI obtains three curves based on C-means, DBSCAN, and DPC algorithms. Each curve points out a suggested number of clusters, which may be different from the others. The optimal number of clusters can be selected by comparing the values of the boundary matching degree of *bou* (*c*).

Figure 12 shows the evaluating process of the multiple-peak dataset that is shown in Figure 2a. It shows that CVIBI based on DBSCAN suggests 2 as the optimal number of clusters, but CVIBI based on C-means and DPC both do 1. To assess their differences, we further analyze their values of *bou* (*c*). *bou* (*c*) = 1 in CVIBI based on C-means and DPC when *c* = 1, while *bou* (*c*) = 1 in CVIBI based on DBSCAN when *c* = 1 and 2. In views of the clustering process, we can conclude that any real cluster in all clusters is not separated when executing clustering algorithm if *bou* (*c*) = 1. Let Γ be the set of numbers of clusters whose *bou* (*c*) = 1. The maximum value in Γ indicates the optimal number of clusters in the multiple-peak dataset. So in the situation, both CVIBI and *bou* (*c*) can indicate that the real number of clusters is 2, and the clustering algorithm suitable for this dataset is DBSCAN.

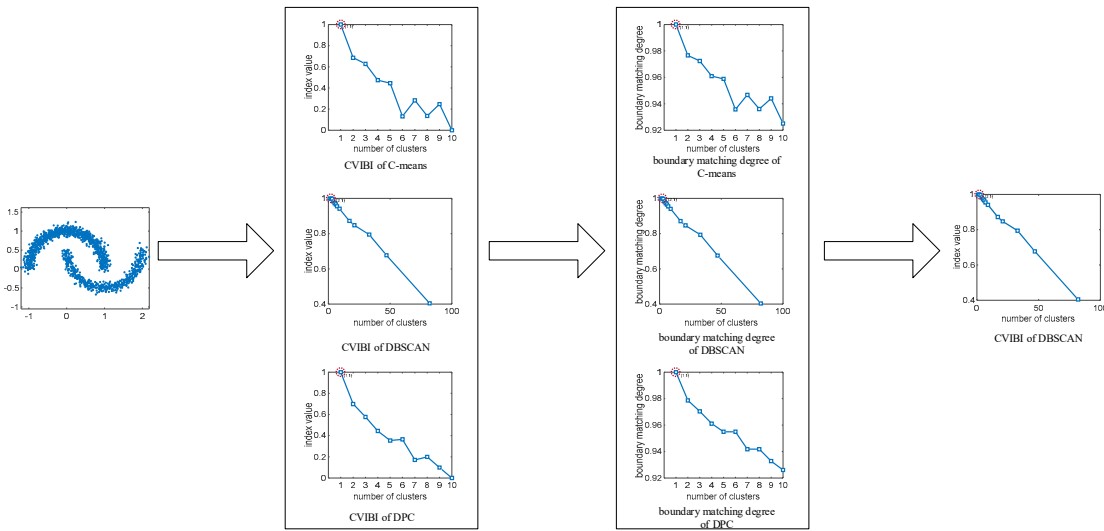

**Figure 12.** Evaluating the process of CVIBI on the multi-peak dataset.

Figure 13 shows the evaluating process of the density-diverse dataset in Figure 2b. We can conclude from Figure 13 that the suggested numbers of clusters are different. CVIBI with DBSCAN and DPC both suggest 2 as the optimal number of clusters while CVIBI based on C-means regards 3 as the optimal number of clusters. To assess which clustering result is best among the three clustering algorithms, the three curves of *bou* (*c*) are used to consist of Γ. Finally, the optimal number of clusters can be determined from Γ. It can be extracted that Γ set contains 1, 2, and 3 in total. So the suggested number of clusters is 3, which is the maximum value in Γ. The optimal algorithm for the density-diverse dataset is C-means algorithm.

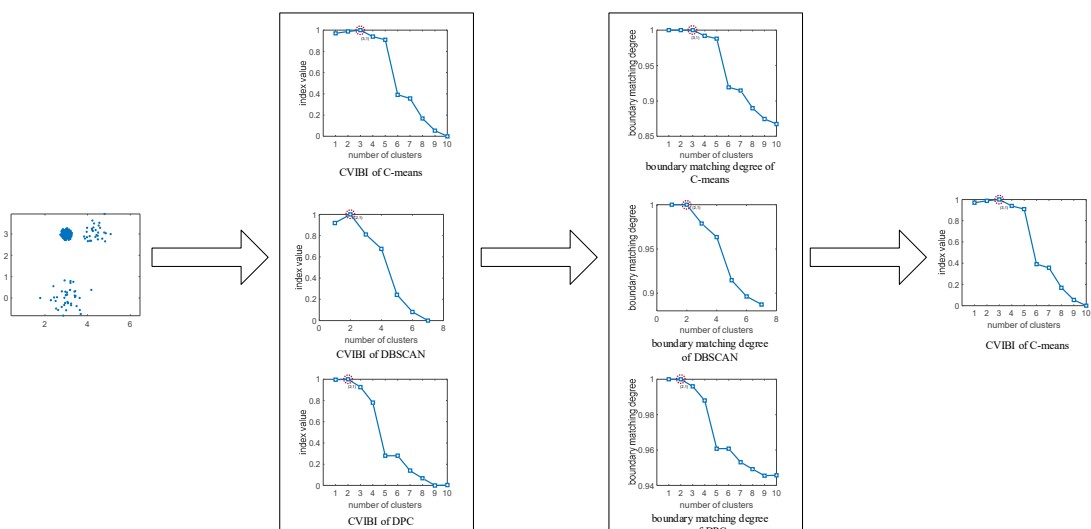

**Figure 13.** Evaluating process of CVIBI on density-diverse dataset.

Figure 14 shows the evaluating process of the dataset in Figure 2c. The results of the CVIBI index with three clustering algorithms are different. CVIBI based on C-means, DBSCAN, and DPC suggest 1, 2, and 3 as the real number of clusters, respectively. In order to obtain the optimal number of clusters, three curves of *bou* (*c*) are used. From the relationship between Γ and the real number of clusters, *bou* (*c*) of C-means, DBSCAN and DPC suggest 1, 2, and 3 as the optimal number of clusters, respectively. So the real number of clusters is 3, which is suggested by CVIBI with DPC. The suitable clustering algorithm of the investigate dataset is the DPC algorithm.

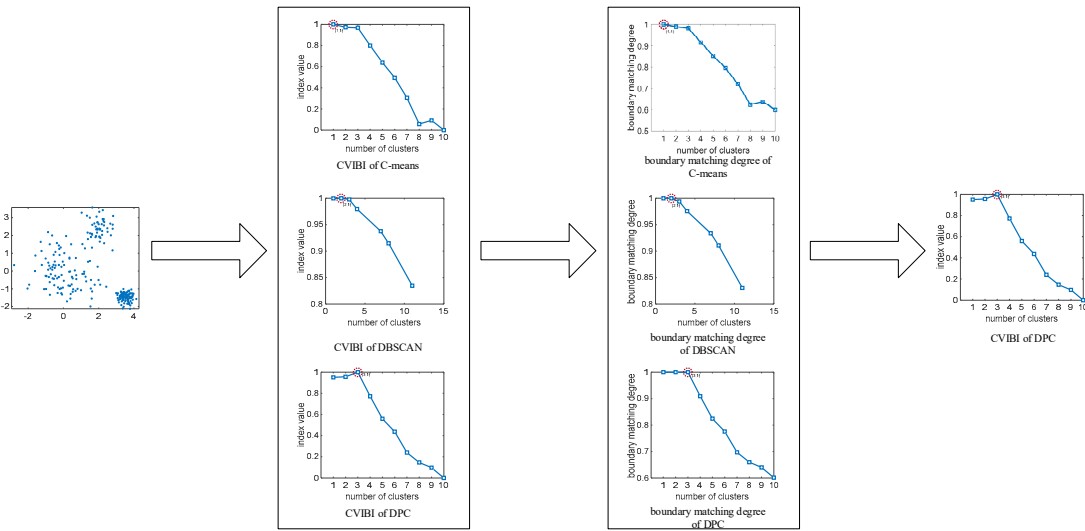

**Figure 14.** Evaluating the process of CVIBI on the overlapped dataset.

### 4.1.4. Comparison between CVIBI and Existing Validity Indices

Table 3 shows all evaluation results obtained by CVIBI, DB, DC, and GS when using the above three clustering algorithms. The four validity indices are analyzed as follows.

(1) Size and distribution. Groups 2 and 4 denote clusters with different sizes and distributions, and there are no overlapped clusters. The evaluation results show that except set 10, all validity indices are capable of determining the correct number of clusters no matter which clustering algorithm is used.

(2) Density and shape. Groups 1 and 3 contain clusters with different densities and shapes. When the densities among clusters are so skewed as in sets 1 and 3, the three validity indices cannot find the correct number of clusters. Compared with DC and DB, the cluster numbers calculated by CVIBI are closer to the real numbers of clusters.

(3) Overlap. Group 5 contains datasets with overlapped clusters. DPC is good at clustering such datasets, and the evaluation results of CVIBI are nearest to the real number of clusters. However, DC and GS with DPC give a relatively smaller number. Note the two indices are originally designed to evaluate the results for nonoverlapping clusters. Thus any two overlapped clusters may be incorrectly regarded as one cluster. For instance, the clusters in set 14 are most overlapped, and its evaluation results are the weakest. How to find the correct number of clusters in a dataset containing overlapped clusters is a difficult task for most existing validity indices, such as DB and GS; fortunately, CVIBI with DPC is effective for dealing with such a problem. In summary, DC shows performances better than CVIBI in the case of datasets containing spherical clusters.

**Table 3.** Evaluation of clustering results by CVIBI, Davies–Bouldin (DB), dual-center (DC) and gap statistic (GS) for 15 datasets.

| Dataset and Algorithm | DB | | | DC | | | GS | | | CVIBI | | |
|---|---|---|---|---|---|---|---|---|---|---|---|---|
| | C-Means | DBSCAN | DPC | C-Means | DBSCAN | DPC | C-Means | DBSCAN | DPC | C-Means | DBSCAN | DPC |
| Set 1 | 3√ | 2 | 3√ | 3√ | 2 | 3√ | 3√ | 2 | 3√ | 3√ | 3√ | 3√ |
| Set 2 | 4√ | 3 | 4√ | 4√ | 3 | 4√ | 4√ | 2 | 4√ | 4√ | 4√ | 4√ |
| Set 3 | 4 | 6 | 4 | 4 | 4 | 5√ | 3 | 3 | 5√ | 5√ | 5√ | 5√ |
| Set 4 | 3√ | 3√ | 3√ | 3√ | 3√ | 3√ | 3√ | 3√ | 3√ | 3√ | 3√ | 3√ |
| Set 5 | 4√ | 4√ | 4√ | 4√ | 4√ | 4√ | 4√ | 4√ | 4√ | 4√ | 4√ | 4√ |
| Set 6 | 4 | 5√ | 4 | 5√ | 4 | 5√ | 4 | 5√ | 5√ | 5√ | 5√ | 5√ |
| Set 7 | 2 | 2 | 2 | 2 | 2 | 2 | 2 | 2 | 2 | 2 | 3√ | 3√ |
| Set 8 | 2 | 4√ | 2 | 4√ | 2 | 4√ | 2 | 2 | 4√ | 2 | 4√ | 4√ |
| Set 9 | 2 | 5√ | 2 | 5√ | 2 | 5√ | 2 | 2 | 5√ | 2 | 5√ | 5√ |
| Set 10 | 2 | 2 | 2 | 2 | 2 | 2 | 2 | 2 | 2 | 2 | 2 | 3√ |
| Set 11 | 4√ | 4√ | 4√ | 4√ | 4√ | 4√ | 4√ | 4√ | 4√ | 4√ | 4√ | 4√ |
| Set 12 | 5√ | 5√ | 5√ | 5√ | 5√ | 5√ | 5√ | 5√ | 5√ | 5√ | 5√ | 5√ |
| Set 13 | 2 | 1 | 2 | 1 | 2 | 1 | 2 | 2 | 1 | 2 | 1 | 2 |
| Set 14 | 2 | 2 | 2 | 2 | 2 | 2 | 2 | 2 | 2 | 2 | 3 | 2 |
| Set 15 | 2 | 1 | 2 | 1 | 2 | 1 | 2 | 2 | 1 | 2 | 1 | 4 |

Note: for the sign 'Set x/y', x and y refer to the investigated dataset and the correct number of clusters, respectively.

### 4.2. Tests on CT Images

Digital Imaging and Communications in Medicine (DICOM) is a standard protocol for the management and transmission of medical images, which is widely used in healthcare facilities. Here, we choose CT images as test datasets, which are stored in accordance with the DICOM standard. Each image contains $512 \times 512$ pixels, and each pixel is identified by its CT value. Figures 15 and 16 show the clustering and evaluation results of CVIBI for a group of typical CT images. The first column represents the original CT images. Columns 2–4 represent the partition results, which are clustered by DPC and C-means, respectively. Here, pseudo colors denote different clusters. The fifth column is the curves of CVIBI.

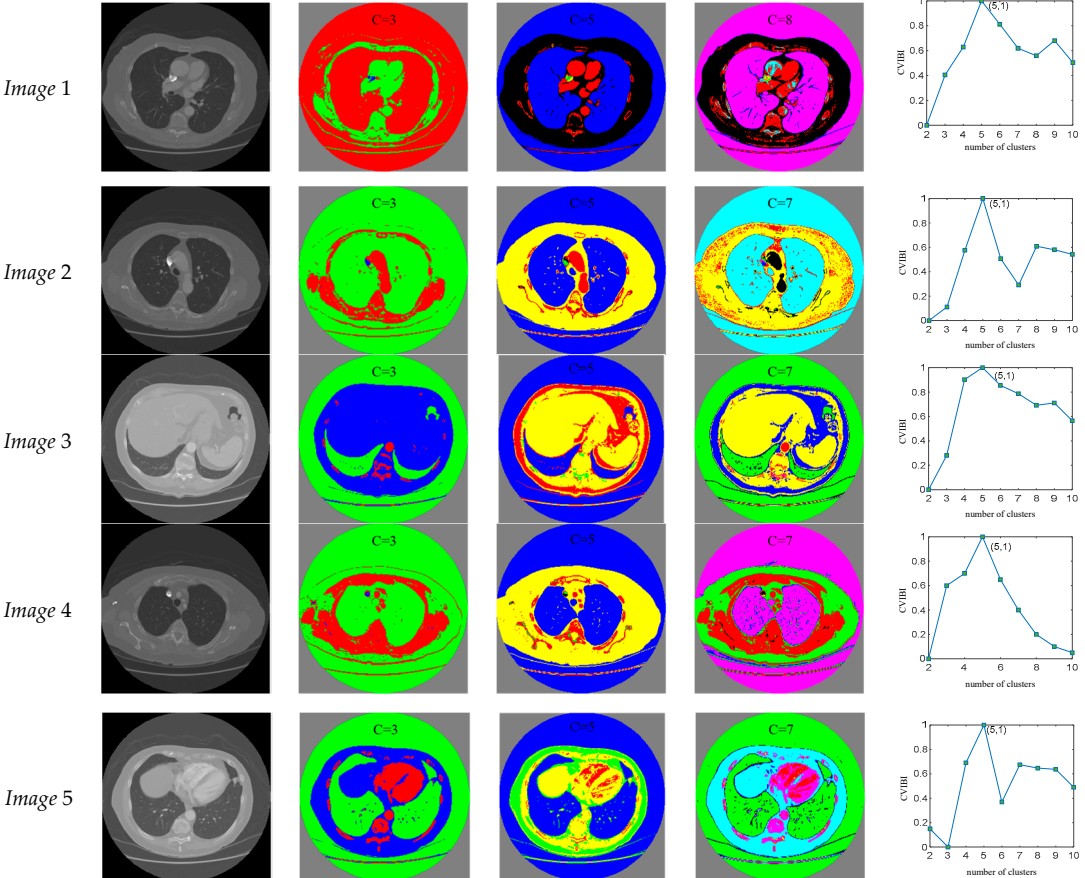

**Figure 15.** Tests of CVIBI on CT images with C-means.

Figures 15 and 16 show that we can obtain the optimal number of clusters when applying CVIBI with C-means and DPC to CT images, and the partitioned images show the shapes of various origins and tissues. Consequently, CVIBI with any clustering algorithms can take effect on automatic imaging segmentation in CT images, which can point out the categories of tissue in one particular CT layer.

Table 4 shows all evaluation results by using C-means and DPC with four indices, respectively, where $x$ and $y$ in sign image $x/y$ refer to the investigated CT images and the correct number of clusters, respectively. The suggested numbers of clusters between the four validity indices are different; GS can identify the correct number of clusters no matter which clustering algorithm is applied; DC with C-means can identify the correct number of clusters, but DC with DPC fails for images 2 and 3; in terms of accuracy, CVIBI seems to be more efficient.

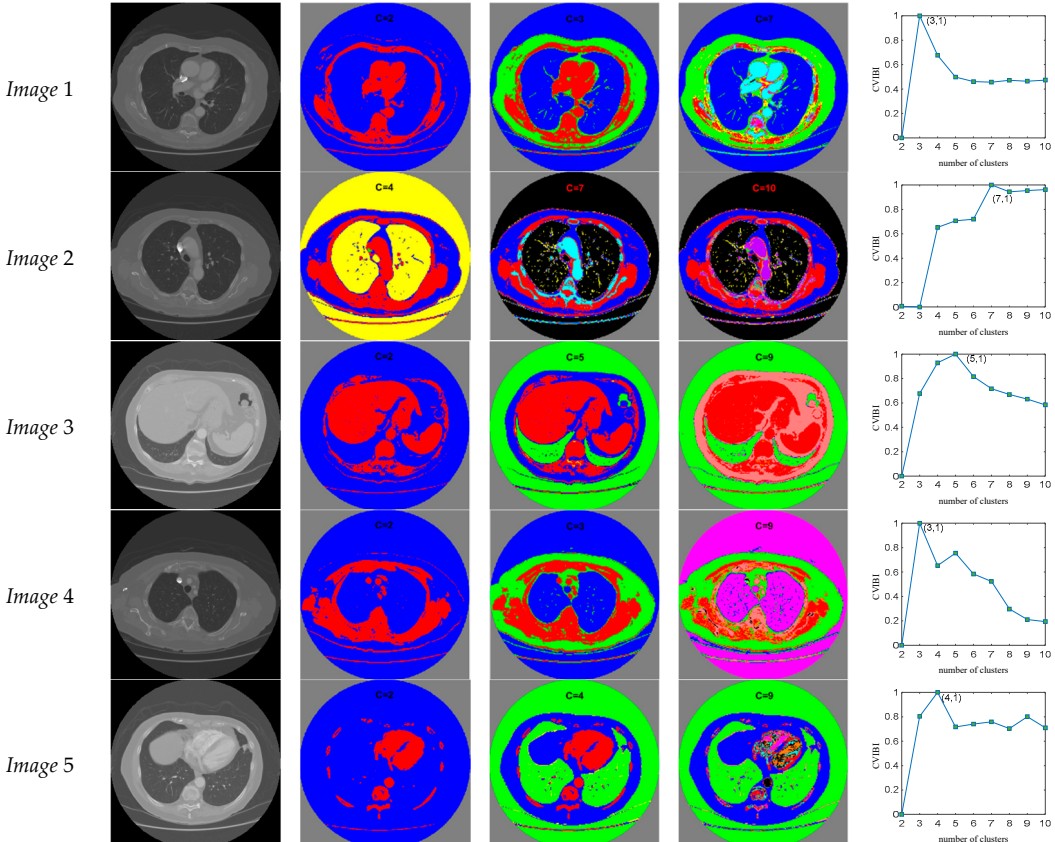

**Figure 16.** Tests of CVIBI on CT images with DPC.

**Table 4.** Evaluation of clustering results by CVIBI, DB, DC and GS for 15 datasets.

| Dataset and Algorithm | DB | | DC | | GS | | CVIBI | |
|---|---|---|---|---|---|---|---|---|
| | DPC | C-Means | DPC | C-Means | DPC | C-Means | DPC | C-Means |
| Set 1 | 4 | 5√ | 3√ | 5√ | 3√ | 5√ | 3√ | 5√ |
| Set 2 | 2 | 3 | 7√ | 3 | 7√ | 5√ | 7√ | 5√ |
| Set 3 | 2 | 4 | 5√ | 4 | 5√ | 5√ | 5√ | 5√ |
| Set 4 | 2 | 5√ | 3√ | 5√ | 3√ | 5√ | 3√ | 5√ |
| Set 5 | 4√ | 5√ | 4√ | 5√ | 4√ | 5√ | 4√ | 5√ |

## 5. Conclusions

The clustering evaluation is an essential but difficult task in clustering analysis. Currently, the existing validity evaluation has to depend on a specific clustering algorithm, a specific cluster parameter (or several), and specific assumptions, and has very limited applicable range. In this paper, we proposed a novel validity index, which can evaluate the clustering results obtained either by a single clustering algorithm or by several clustering algorithms. Especially, it can be applied to select any clustering parameters besides the typical number of clusters. To our knowledge, the kind of necessary applications cannot be realized by existing validity indices. This novel index outperforms the existing validity indices on some benchmark datasets in terms of accuracy and generality. Experimental results validate this index. The boundary matching degree and connectivity degree are important notions in graph theory. Our future work is to combine these notions with graph theory to reduce time complexity.

**Author Contributions:** Conceptualization, Q.L. and S.Y.; methodology, Q.L. and S.Y.; software, Q.L. and Y.W.; validation, Q.L., M.D. and J.L.; formal analysis, Z.W.; investigation, J.L. and Z.W.; resources, Z.W.; data curation,

Q.L. and M.D.; writing—original draft preparation, Q.L.; writing—review and editing, M.D. and J.L.; visualization, Y.W.; supervision, S.Y.; project administration, S.Y.; funding acquisition, S.Y. All authors have read and agreed to the published version of the manuscript.

**Funding:** This research was funded by the National Natural Science Foundation of China, grant number 61573251 and 61973232.

**Conflicts of Interest:** The authors declare no conflict of interest.

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
