# Peer review of "Boundary Matching and Interior Connectivity-Based Cluster Validity Anlysis"

_applsci, doi:10.3390/app10041337_

Round 1
Reviewer 1 Report
The authors present a measure to validate clustering results.
1) There are many problems regarding the use of the English language and I suggest the authors correct the manuscript with the aid of a native English speaker.
2) The Related work section is well Witten. I have however a concern. The authors do not mention any Bayesian approach for validating clustering results. For example the paper below
Teklehaymanot, Freweyni K., Michael Muma, and Abdelhak M. Zoubir. "Bayesian cluster enumeration criterion for unsupervised learning." IEEE Transactions on Signal Processing 66.20 (2018): 5392-5406
I would be good the authors to mention and briefly discuss Bayesian approaches too.
3) In Table 1 there is a typo. You have to use the same symbol to represent cluster centers.
4) I think that there is a typo in equation (7). You should replace x_i with x_k.
5) In lines 222, 223 and 259 you probably refer to relation (10) and not to relation (11).
6) Some of the figures have labels in Chinese. Please correct them. For example Figure 7(d), Figure 14 and Figure 15.
7) Line 361, you mention that you use DBSCAN with all possible values of Eps. If I am correct, Eps is a continuous variable. So, what do you exactly mean by “all possible Eps”?
8) Line 390, you mention that the maximum value in Γ indicates the optimal number of clusters. Intuitively, that makes sense, however, I guess that there might be cases where such an argument is wrong. Please elaborate more.
9) In table 6, the title of the last column is BIC. Is that correct? Are you referring to Bayesian Information Criterion? Or is the title a typo?
Reviewer 2 Report
The manuscript proposes a new clustering evaluation metric more suitable to a generic clustering algorithm, in comparison to most of the existing metrics that are suitable for specific clustering algorithms. The experiments section is thorough and well-explained to show the advantages of the proposed clustering evaluation strategy. Overall, the manuscript shows the promise of the proposal.
There are a few concerns that need to be addressed before the manuscript can be accepted.
There are many grammatical and visualization problems, that require a major proof-reading. Only a few are mentioned next: In abstract, “farther” should be “further”, to make any sense. In fact, the abstract is not well written. The boundary points extraction is actually part of the metric as well (since the nonparametric way of computing boundary points is part of the process). Hence, the authors are advised to rewrite the abstract. In section 3.1, distances are symbolized as d(), but suddenly change to “dis()” in eq. (10). 7 and 15 need to be in English. 13 most likely shows a wrong curve at the right side. Shouldn’t it choose 3 as the optimal number of clusters and DPC as the method? The number of neighbors to compute the density is still a bit confusing. What was the number used in the experiments? Was it fixed throughout the experiment? The concern is: varying this may significantly vary the density, the distances, and subsequently the metric values. A study on this in the manuscript is also important from the point of view of robustness of the measure.Author Response
Please see the attachment.

Round 2
Reviewer 2 Report
The authors incorporated all the suggestions given by the reviewers and considerably improved the manuscript.
There is still one major problem that needs to be fixed before accepting the manuscript. Although it is a major problem, it needs a minor correction, and hence, needs a minor revision at this time.
Here's the problem: the authors have corrected the confusion between dis() and d() as distance between neighboring data points. However, in response to reviewer's request, they explained the optimal number of neighbors as the integer part of "2dπ" (read - 2 * d * pi). The manuscript required another careful read to understand that "d" referred to the "dimension" of the data, and not the distance d().
Firstly, the authors need to confirm whether it is correct, and confirm that due to all experimental datasets used in the study being two dimensional (according to Table 4), the number of neighbors used for all data are: [2 * 3.14159265358... * 2] = 12 (from the graph in Fig. 9, it is not clear if the number is exactly 12).
Secondly, they need to change the symbol of either distance d() or dimension d, in order to clear this major confusion. My suggestion would be to use capital D for dimension.
